

# Differential p16 expression levels in the liver, hepatocytes and hepatocellular cell lines

Barbara Kramar[1], Dušan Šuput[1] and Irina Milisav[1,2]

[1] University of Ljubljana, Faculty of Medicine, Institute of Pathophysiology, Zaloska 4, Ljubljana, Slovenia
[2] University of Ljubljana, Laboratory of oxidative stress research, Faculty of Health Sciences, Zdravstvena pot 5, Ljubljana, Slovenia

## ABSTRACT

**Background:** One of the most frequently deleted genes in cancer is *CDKN2A* encoding p16. This protein is often overexpressed in senescent cells, while its suppression can bypass the oncogene-induced senescence to enable transformation and tumorigenesis. The roles of the protein p16 are recently being expanded from the cell cycle progression regulator to the cellular regulator interacting in several different pathways. Yet data on its liver and liver cells' expression are inconclusive.
**Methods:** The expression of the *p16* gene in liver and liver cells was determined by RT-qPCR and compared to its protein amounts by western blotting.
**Results:** *p16* is expressed at low levels in the liver and rat hepatocytes. Its expression varies from none to the considerable levels in the examined hepatocellular carcinoma cell lines (FaO and HepG2) and in immortalized mouse hepatocytes. Such significant expression differences of an important cellular regulator warrant the need to closely examine the differences in biochemical pathways correlated with the *p16* expression when using hepatocytes and hepatoma liver models.

## INTRODUCTION

The cyclin-dependent kinase inhibitor 2A (*CDKN2A*) gene encodes two protein regulators of cell cycle regulatory pathways, the *p16*(*INK4A* or *p16*) and *p14*(*ARF*), which are encoded in alternative reading frames (*Stone et al., 1995*). Protein p16 is a regulator of cellular homeostasis controlling growth, senescence, apoptosis and cellular differentiation. It inhibits cell cycle progression through the cyclin-dependent kinase-4 and -6/retinoblastoma (CDK4/6/Rb) pathway (*Tyagi et al., 2017*). It is an inhibitor of a cyclin-dependent kinase-4 (CDK4) (*Angelico et al., 2021*). This inhibition maintains the retinoblastoma protein (pRb, a product from the Rb gene) in a hypo-phosphorylated state and prevents the cell cycle progression from G1 to S phase (*Li, Poi & Tsai, 2011*).

In addition to this canonical pathway, several retinoblastoma-independent pathways of p16 have been described recently. The examples include (1) regulation of nucleotide biosynthesis involving a pentose phosphate pathway enzyme ribose five-phosphate isomerase A (*Buj et al., 2019*). (2) A direct interaction of p16 with the p65 subunit of the

Corresponding author
Irina Milisav,
irina.milisav@mf.uni-lj.si

NF-κB complex can decrease the NF-κB-promoted tumorigenesis in the absence of the inhibitor IκBα (*Wolff & Naumann, 1999*). (3) p16 can bind and inhibit JNK1/3 by binding to the c-jun binding site to impair the activation of AP-1 (*Choi et al., 2005*). (4) It also impairs the transcription factor eEF1A2 translational activity (*Lee et al., 2013*). Further functions involve (5) oxidative stress decrease (*Jenkins et al., 2011*) and (6) mitochondrial biogenesis (*Tyagi et al., 2017*).

Protein p16 is expressed in normal tissues and in solid tumours (*Angelico et al., 2021*). Its non-cancerous tissue expression is mostly low to moderate according to the Human Protein Atlas data (*Atlas*) (CDKN2A), with high expression only in glandular cells of the colon and in late spermatids. Nevertheless, the reduced amounts of p16 tumour suppressor enable transformation and tumorigenesis (*Buj et al., 2019*). Therefore, its role has been mostly investigated in cancer and ageing (*Buj & Aird, 2019*). p16 reduction is linked to different types of cancers, *e.g.* melanoma, lymphoma, pancreatic adenocarcinoma, non-small cell lung cancer, gastric cancer, colorectal cancer, etc. (*Zhao et al., 2016*).

Loss of p16 expression because of promoter methylation was reported in 40% of the investigated cell lines in one study (*Murai et al., 2005*) and it also commonly occurs in hepatocellular carcinoma (HCC) (*Lv et al., 2017*). This is a prevalent malignancy in Asia (*Lv et al., 2017*), the fifth most common malignant disease in men worldwide and the cause of the second most common cancer-related deaths in men (*Torre et al., 2015*). The authors of a meta-analysis reported that p16 promoter methylation in HCC was increased with age and hepatitis virus B and C infections (*Lv et al., 2017*). Nevertheless, p16 can be expressed or not in HCC and examination of the p16 status along with another tumour suppressor p27 was proposed as a more accurate tool for predicting the prognosis of HCC (*Matsuda & Ichida, 2006*). Other intragenic mutations of the p16 gene were also reported in HCC in addition to the promoter methylation (*Kita et al., 1996*). No p16 expression is reported for human hepatocytes so far (*Atlas*)(CDKN2A), although p16 was reported as one of the senescent markers in old mice on a high-fat diet (*Ogrodnik et al., 2017*). Low liver expression of p16 in human liver cholangiocytes, but not in hepatocytes, is described (*Atlas*)(CDKN2A) and a p16 hepatocyte expression was detected by immunohistochemistry in the cases of advanced liver fibrosis (*Csepregi et al., 2008*).

Here we describe the differences in p16 expression levels among liver, primary and immortalized hepatocytes and hepatoma cell lines and discuss the implications of expression differences of this potentially important regulator of several cellular pathways.

## MATERIALS & METHODS

All reagents were purchased from Sigma-Aldrich (Merck), unless otherwise stated.

### Ethical statement

Primary hepatocytes and the liver were isolated from adult male rats (Wistar-Hannover, Ljubljana, Slovenia, 180–280 g), ethical code numbers are U34401-44/2014/8 and U34401-21/2020/4, issued by Administration of the Republic of Slovenia for Food Safety, Veterinary Sector and Plant Protection.

## Cell cultures

All cell models were grown at 37 °C in a humidified atmosphere with 5% $CO_2$. The rat hepatoma cell line FaO (ECACC, 89042701) was grown as previously described (*Pirc Marolt et al., 2020*). They were seeded in six-well cell culture plates (650.000 cells/well) and grown for 48 hours before harvest.

The human hepatoma cell line HepG2 (ATCC, HB-8065) was grown in Dulbecco's Modified Eagle's Medium (DMEM; Gibco, 11966) with 2 g/L D-(+)-glucose (G6152), supplemented with 10% FBS and 1% Pen-Strep for 48 hours before harvest. Six-well cell culture plates (500.000 cells/well) were used for the seeding.

The human embryonic kidney cell line 293T (ECACC, 12022001) is a single-cell clone of regular 293 cells. The cells were seeded in six-well cell culture plates (200.000 cells/well) and grown in high glucose DMEM (4.5 g/L glucose + L-glutamine, 11965–092), 10% FBS and 1% Pen-Strep for 48 hours before harvest.

Immortalized mouse neonatal hepatocytes provided by Dr Angela M. Valverde (*Gonzalez-Rodriguez et al., 2009*; *Pardo et al., 2015*) were grown in T25 cell culture flasks (seeded 100.000 cells/flask) in a high glucose DMEM (4.5 g/L glucose + L-glutamine, 11965–092), 10% FBS and 1% Pen-Strep for 48 hours before harvest.

Liver from 12-week old Wistar rats and primary hepatocytes from 8-week old Wistar-Hannover rats were isolated under ethical code numbers stated above. Reverse two-step perfusion with collagenase (C5138) was used to isolate primary hepatocytes (*Nipic et al., 2010*). Hepatocyte viability was at least 90%, as determined by Trypan blue stain 0.4% (Gibco, 15250). The cells were seeded on the collagen type I (C867) coated surface of six well plates at 500.000 cells/well for 4 hours in Williams medium E (W4125) supplemented with 10% FBS, 1% Pen-Strep and insulin (0.1 U/mL; I1882). For the remaining 68 hours until harvest, hepatocytes were in the Williams medium E with 0.03% bovine serum albumin (A2153), 0.5% Pen-Strep, insulin (0.1 U/mL) and 1 μM hydrocortisone-21 hemisuccinate (H2270). The liver were processed as previously described (*Banic et al., 2011*).

## RNA isolation and reverse-transcription quantitative polymerase chain reaction analysis (RT-qPCR)

Total RNA was isolated with TRI reagent (T9424) and reverse transcribed using the High capacity cDNA reverse transcription kit (4368814; Applied Biosystems) with added RNase inhibitor (N8080119; Applied Biosystems). PCR reactions (≤100 ng cDNA/ reaction) were run in duplicates using TaqMan Universal Master Mix II, with uracil-N-glycosylase (4440038; Thermo Fisher Scientific) and quantitated using the 7,500 Real-Time PCR System with SDS software (v1.3.1, Applied Biosystems) or on QuantStudio three Real-Time PCR System with Design and Analysis 2.5.0. software (Thermo Fisher Scientific). The software was used to set the baseline (auto) and to determine the cycle threshold (Ct). The following TaqMan probes labelled with the FAM dye (Thermo Fisher Scientific) were used: rat p16 (Rn00580664_m1), mouse p16 (Mm00494449_m1) and human p16 (Hs00923894_m1). 18S (Hs99999901_s1) was used as a reference gene.

## Immunoblotting

Whole cell lysates in a lysis buffer (20 mM Hepes-KOH (H3375)), pH 7.9, 125 mM NaCl (Fluka, 31434), 1 mM EDTA (E6758), 1% Igepal CA-360 (I8896), 10 mM sodium pyrophosphate (P8010), 5 mM sodium fluoride (S6776), 5 mM β-glycerol phosphate (50020), 0.2 mM sodium orthovanadate (450243), 1mM phenylmethylsulfonyl fluoride (PMSF, P7626) and 1 mM protease inhibitor cocktail (P8340) were added to the reducing Laemmli-buffer (0.25 M Tris pH 6.8, 8% SDS, 40% glycerol, 0.03% bromophenol blue), denatured at 95 °C for 5 min and 40 μg of a sample was applied on 16% (w/v) acrylamide gels to be separated by standard sodium dodecyl sulfate–polyacrylamide gel electrophoresis (SDS-PAGE) and blotted onto PVDF membrane (Immobilon-P; Merck-Millipore, Darmstadt, Germany). Two antibodies against p16 were used: the rabbit polyclonal (PA5-20379; Invitrogen, Thermo Fisher Scientific, Waltham, MA, USA) and the mouse monoclonal Anti-CDKN2A/p16INK4a Antibody (F-12; sc-1661, Santa Cruz Biotechnology Inc., Dallas, TX, USA). The signal was detected by luminescence through the secondary goat anti-rabbit and anti-mouse antibodies, respectively, conjugated to horseradish peroxidase (Bio-Rad, Hercules, CA, USA), visualized using Fusion FX imager (Vilber, Marne-la-Vallée, France) and quantified by densitometry using Image Studio Lite software (LI-COR, Lincoln, NE, USA).

## Statistical analysis

All data are presented as means ±SD. The number of biological replicates is stated in the figure legends. GraphPad Prism 9.1.2 with an inbuilt algorithm to test the equality of variances from medians, the Brown-Forsythe test, was used for statistical analysis. In the case of equal variances, one-way ANOVA was used and Tukey's multiple comparisons test.

## RESULTS

Expression of the *p16* gene was determined in rat liver, hepatocytes and hepatocyte-derived cells (Fig. 1) and compared to the expression in HEK 293T cell line with reported *p16* expression (*Cui et al., 2015*). Low levels of *p16* expression were detected in the liver and primary rat hepatocytes. Somewhat more transcript was in HepG2 cells. Comparable amounts of *p16* were transcribed from immortalized hepatocytes and HEK 293T cell line (Table 1). In contrast to the *p16* expression in HepG2, there was no expression whatsoever in the FaO hepatoma cell line.

The expression of 18S RNA served as a cDNA quality control. The mean threshold cycle (Ct) values of the reference gene are similar; the arithmetic mean +/- SD is 14.39 +/- 0.29. The histograms of delta Ct (ΔCt) values, therefore, follow an identical pattern to that of the Ct values with statistically significant differences between all hepatocyte cell lines (Fig. 2).

In conclusion, similar gene expression was measured in primary hepatocytes of young (pre-sexually matured) rats and in the liver of adult rats. The *p16* expression markedly differed among the two hepatoma cell lines and immortalized hepatocytes, in which the expression ranged from 0 to well expressed, with the Ct number around 24 and ΔCt around 10 (Fig. 2).

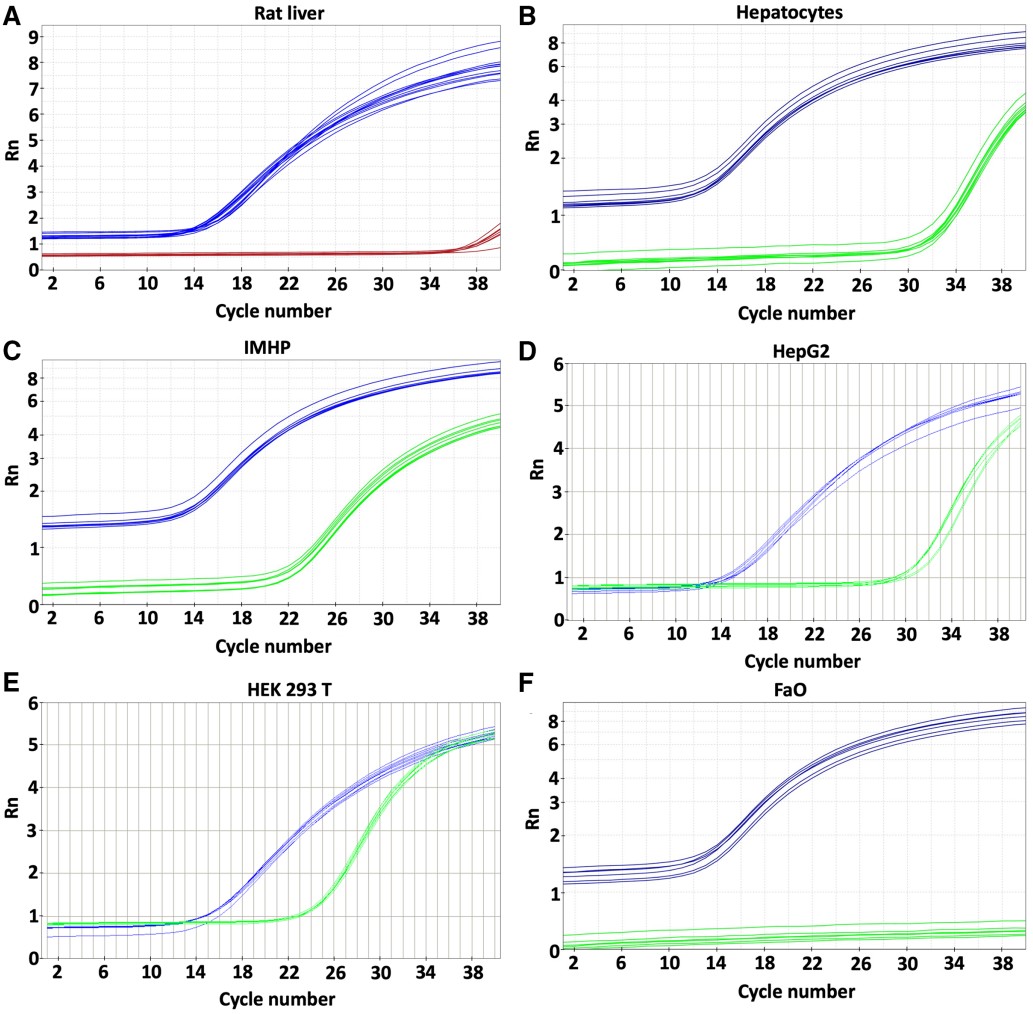

**Figure 1 Gene expression.** *p16* gene was determined in rat liver (A) and cellular liver models: primary rat hepatocytes (B), immortalized mouse hepatocytes (IMHP) (C), human HepG2 cell line (D), Human embryonic kidney 293T (HEK 293T) cell line (E) and rat hepatoma FaO cell line (F). Rn: normalized reporter value (fluorescence of the reporter dye/fluorescence of a passive reference dye). *18S*: blue, *p16*: red/green.          

Gene expression data were then compared with the protein levels. Amounts of p16 protein were visualized by two commercial antibodies, a polyclonal (Fig. 3) and a monoclonal (Fig. S1). Both detected many bands of different sizes in all tested liver cells and tissue, however, the bands running close to a 17 kDa protein marker closely resembled a DNA expression pattern. The relative differences in protein amounts between the hepatic cells and tissue follow the gene expression pattern of large statistically significant differences (Fig. 3).

## DISCUSSION

We found large differences in expression levels of the *p16* gene in the liver and various closely related hepatic cells that are widely used as experimental models. Unlike the expression levels of a reference gene that was stable, the expression of *p16* ranged from zero

**Table 1 Mean threshold cycle (Ct) values and standard deviations of biological replicates of tested liver, hepatocytes and cell lines.**

| Cell model | Mean Ct values | SD | P values (Tukey's multiple comparisons test) | | | |
|---|---|---|---|---|---|---|
| | | | IMHP | HepG2 | Hepatocytes | Rat liver |
| HEK 293T | 24.74 | 0.127 | 0.064 | $4.2 \times 10^{-12}$ | $2.5 \times 10^{-14}$ | $2.3 \times 10^{-14}$ |
| IMHP | 23.91 | 0.146 | / | $6.3 \times 10^{-13}$ | $2.3 \times 10^{-14}$ | $2.3 \times 10^{-14}$ |
| HepG2 | 31.07 | 0.544 | / | / | $6.5 \times 10^{-6}$ | $3.5 \times 10^{-9}$ |
| Hepatocytes | 33.41 | 0.237 | / | / | / | $6.3 \times 10^{-5}$ |
| Rat liver | 35.36 | 0.853 | / | / | / | / |
| FaO | No amplification | | – | – | – | – |

Note:
Human embryonic kidney (HEK 293T), immortalized mouse hepatocytes (IMHP), primary rat hepatocytes, $n = 4$; rat liver, HepG2 and FaO cell line $n = 3$. The differences between the cell lines are statistically significant ($P < 0.1 \times 10^{-14}$, one-way ANOVA; normal distribution of data: Brown-Forsythe test $P = 0.47$).

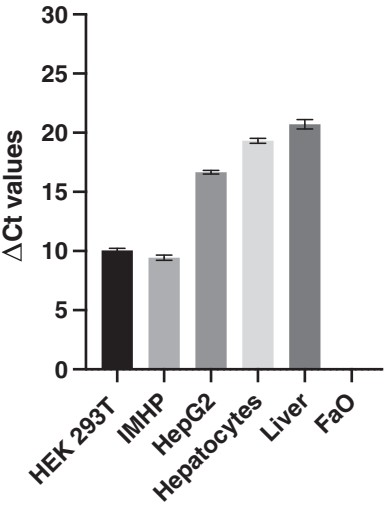

**Figure 2 Delta Ct values.** Mean delta Ct values of *p16* and *18s RNA* amplification in liver, hepatocytes and cell lines. Human embryonic kidney 293T (HEK 293T) cell line, immortalized mouse hepatocytes (IMHP), primary rat hepatocytes, $n = 4$. Rat liver, HepG2 cell line, FaO cell line, $n = 3$. Data are presented as the mean ± standard deviation (SD) and are statistically significant ($P < 0.1 \times 10^{-14}$, one-way ANOVA; normal distribution of data: Brown-Forsythe test $P = 0.56$). Pairwise differences between HEK293T and IMHP: $P = 0.014$; between HEK293T and any other cell type $P < 0.1 \times 10^{-14}$; IMHP and HepG2/hepatocytes/rat liver: $P < 0.1 \times 10^{-14}$; HepG2 and hepatocytes $P = 1.3 \times 10^{-9}$; HepG2 and rat liver $P = 9.2 \times 10^{-12}$; hepatocytes and rat liver $P = 7.3 \times 10^{-6}$.

to substantial levels. The protein amounts of positive control and hepatic cell lines and tissue closely resemble the gene expression levels when the bands that run close to the 17 kDa protein marker are compared. No strong p16 band was observed, which is in line with the Protein Atlas antibody data (*Atlas*). Many bands of different sizes are highlighted when the blots are probed with a polyclonal or monoclonal antibody, which together with a weak signal warrant for the development of an antibody suitable for detection of baseline levels of p16. Post-transcriptional modifications of p16 may account for a band shift in IMHP. Indeed, a p16 protein can be phosphorylated at four sites and

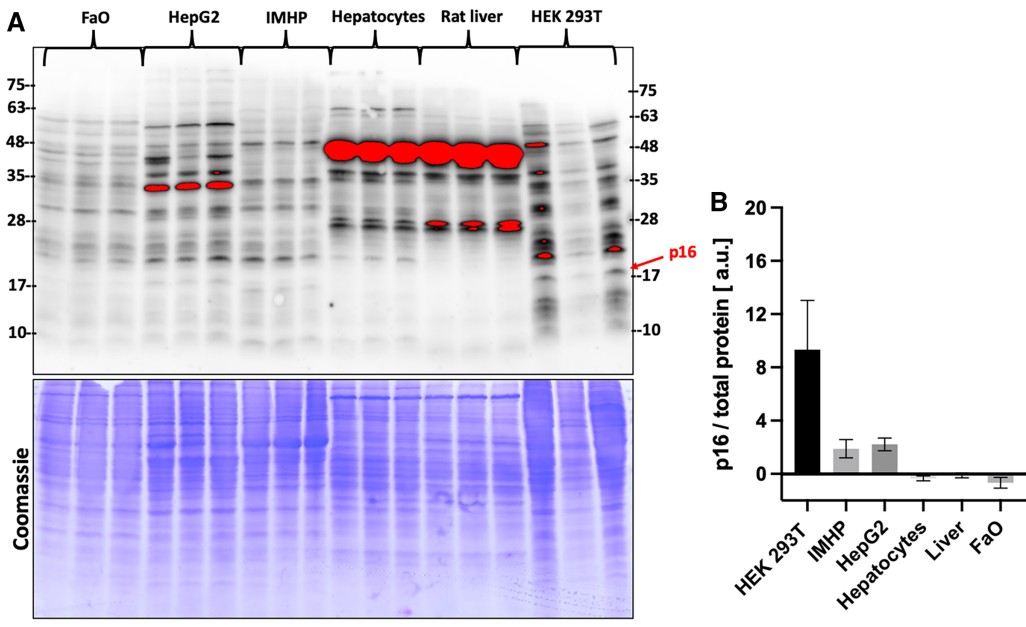

**Figure 3 p16 protein levels.** (A) Immunoblot was probed with a polyclonal antibody to p16. Each lane represents a lysate from one biological replicate. Underneath is a Coomassie-stained gel for loading control. (B) Densitometric analysis of protein levels normalized to a Coomassie-stained gel. The differences in p16 protein amounts are statistically significant ($P = 4.0 \times 10^{-5}$, one-way ANOVA; normal distribution of data: Brown-Forsythe test $P = 0.19$). Pairwise comparison of the protein amounts: HEK293T to IMHP: $P = 8.9 \times 10^{-4}$, to HepG2: $P = 1.3 \times 10^{-3}$, to hepatocytes: $P = 7.5 \times 10^{-5}$, to rat liver: $P = 9.5 \times 10^{-5}$, to FaO: $P = 5.4 \times 10^{-5}$.

also methylated (*Gump, Stokoe & McCormick, 2003*; *Jiao, Feng & Wang, 2018*). The appropriate size of protein detected in the positive control and hepatic samples as well as the relative amounts of various hepatic cells and the tissue are in line with the gene expression data and therefore strengthen the protein band identification. This process also highlights difficulties in comparing the gene expression with the protein data reported by many authors (*Csardi et al., 2015*; *Liu, Beyer & Aebersold, 2016*; *Pascal et al., 2008*).

No p16 protein has been detected in hepatocytes according to the Protein Atlas (*Atlas*) (CDKN2A). Nevertheless, p16 has a role in hepatocellular carcinoma and steatosis progression (see below). Overexpression of *p16* gene was reported to inhibit proliferation and reduce invasion ability of hepatocellular carcinoma (*Huang et al., 2003*). Therefore, these differences in expression imply a need to evaluate *p16* expression in every experimental liver model to establish whether metabolic differences in related cell models are due to variations in *p16* expression.

A correlation was found between the p16 expression and hepatic fat accumulation since the removal of p16 expressing senescent cells in a mice model reduced steatosis (*Ogrodnik et al., 2017*). In approximately 20–30% of people liver steatosis progresses to a harmful nonalcoholic steatohepatitis characterized by liver inflammation, dysfunctional fibrosis, and hepatocyte death (*Rinella, 2015*). No p16 was detected by immunohistochemistry in human non-cancer liver tissue without fibrosis, while there was a significant p16 expression in the case of advanced liver fibrosis (*Csepregi et al., 2008*). Liver fibrosis

develops as a result of hepatic stellate cells (HSCs) activation to myofibroblasts (MFBs), which deposit collagen during hepatic fibrogenesis also to replace the apoptotic hepatocytes (*Sancho et al., 2012*). Transforming growth factor-beta (TGF-β) level increases during the development of liver fibrosis and induces apoptosis in hepatocytes, while also contributes to the activation of HSCs (*Proell et al., 2007*; *Sanchez et al., 1996*).

Escaping from the TGF-β suppression is also a prerequisite for liver tumour progression, as TGF-β also activates the survival signals (*Caja et al., 2007*). This process is important in liver carcinogenesis. TGF-β was reported as a suppressor factor for adult quiescent hepatocytes, but not for FaO hepatoma cells, where it had two roles, both suppressing and promoting carcinogenesis (*Caja et al., 2007*). These are the two cell models in which we have determined a different expression of *p16*. It fits with the notion that there is an overexpression of p16 in senescent cells, while its suppression can bypass the oncogene-induced senescence to enable transformation and tumorigenesis (*Buj et al., 2019*).

## CONCLUSIONS

As p16 can regulate several pathways and is differentially expressed in closely related hepatocytes and hepatocyte-derived cells, it is necessary to check for the possible differences in biochemical pathways in hepatocyte liver models that may arise because of the differences in p16 expression.

## ACKNOWLEDGEMENTS

Izak Patrik Miller and Patrik Prša kindly assisted with liver processing. We are grateful to Dr. Ángela M. Valverde, Instituto de Investigaciones Biomedicas 'Alberto Sols', CSIC, Madrid, Spain for a kind donation of mouse immortalized hepatocytes and to Dr. Maria Monsalve for the HEK 293T cells through the H2020-MSCA-ITN:721236 TREATMENT project.

### Funding

This research was funded by H2020-MSCA-ITN:721236 TREATMENT project and from the Slovenian Research Agency (research core funding No. P3-0019). BK acknowledges the financial support of the Word Federation of Scientists within the framework of the Slovenian Science Foundation. The funders had no role in study design, data collection and analysis, decision to publish, or preparation of the manuscript.

### Grant Disclosures

The following grant information was disclosed by the authors:
H2020-MSCA-ITN:721236 TREATMENT.
Slovenian Research Agency: P3-0019.
Slovenian Science Foundation.

## Competing Interests

The authors declare that they have no competing interests.

## Author Contributions

- Barbara Kramar conceived and designed the experiments, performed the experiments, analyzed the data, prepared figures and/or tables, and approved the final draft.
- Dušan Šuput authored or reviewed drafts of the paper, and approved the final draft.
- Irina Milisav conceived and designed the experiments, analyzed the data, prepared figures and/or tables, authored or reviewed drafts of the paper, and approved the final draft.

## Ethics

The following information was supplied relating to ethical approvals (*i.e.*, approving body and any reference numbers):

Administration of the Republic of Slovenia for Food Safety, Veterinary Sector and Plant Protection approved the study (U34401-44/2014/8 and U34401-21/2020/4).

## Data Availability

The raw data is available at the OpenAIR repository: KRAMAR, Barbara, ŠUPUT, Dušan in MILISAV, Irina, 2021, p16 expression in the liver and liver cells. 2021. https://repozitorij.uni-lj.si/IzpisGradiva.php?id=128425.

## Supplemental Information

Supplemental information for this article can be found online at http://dx.doi.org/10.7717/peerj.12358#supplemental-information.

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
