# Peer review of "Differential p16 expression levels in the liver, hepatocytes and hepatocellular cell lines"

_PeerJ, doi:10.7717/peerj.12358_

## Round 0.1 · original submission · Major Revisions

Please address the concerns of the reviewers and amend the manuscript accordingly.

Reviewer 1 ·

Basic reporting

The manuscript deals with p16 expression levels in different cell models, aiming to discuss the implications of expression differences. The manuscript contains several items which need to be corrected. The Introduction Section contains relevant information.
Raw data is shared by the authors; however, it seems that Table 1 and Figure 2 present the same data.

Experimental design

The authors used appropriate standard qRT-PCR technique and described the cell culture methods in sufficient detail.

Validity of the findings

The raw data Table contains all necessary values for the calculation of ΔCt. To see the real differences in p16 expression the relative quantification of the gene expression should be presented in the Figure and further discussed.

Besides the descriptive Statistics, the authors should also perform the basic analysis of the obtained values to test for the group differences.

Reviewer 2 ·

Basic reporting

The manuscript entitled "Differential p16 expression levels in the liver,
hepatocytes and hepatocellular cell lines (#63307)" is a well written manuscript.The article explores the expression of the p16 gene in various liver cell lines as well as in liver tumor models using RT-qPCR technique. The literature references are sufficient with respect the field. In the results, they observed that there is different expression of p16 depending on the cell types or liver tissues studied. The authors conclude that the different expressions of p16 in different cell cultures or tumor tissues could perhaps be due to metabolic changes of each cell type studied.
In my opinion, it is necessary to study the expression of the p16 protein in order to be able to correlate it with the expression of the gene in this study.

Experimental design

The study seems relevant to me, as well as the question that the researchers ask themselves. However I believe that Western blot or immunohistochemical studies using antibodies against p16 are necessary to study the expression of the protein in different cell cultures and in liver tumor tissues.

Validity of the findings

I think the study is relevant since p16 is an important protein in the study of fibrosis, senescence and cancer in the liver. However, I believe that the study of the p16 gene has to be complemented with the expression of the protein.

---

## Round 0.2 · accepted · Accept

Thank you for addressing the critiques of the reviewers and for amending your manuscript.

Reviewer 1 ·

Basic reporting

Authors additionally improved the text according to reviewers' comments.

Experimental design

Authors additionally improved the methods according to reviewers' comments.

Validity of the findings

Based on the additional experimental data the findings are discussed further.